# Renal Abscess Associated with SGLT2 Inhibitors Administration in Heart Failure Without Other Previous Risk Factors: A Case Report

**DOI:** 10.3390/biomedicines13020389

**Published:** 2025-02-06

**Authors:** Madalina Andreea Munteanu, Camelia Nicolae, Razvan Ionut Popescu, Andreea Rusescu, Nicolae Paun, Tiberiu Ioan Nanea

**Affiliations:** 1Internal Medicine and Cardiology Department, Carol Davila University of Medicine and Pharmacy, 020021 Bucharest, Romania; dr.amunteanu@gmail.com (M.A.M.); nicpaun66.np@gmail.com (N.P.); tiberiu.nanea@yahoo.com (T.I.N.); 2Cardiology and Urology Department, Clinical Hospital “Prof. Dr. Th. Burghele”, 050659 Bucharest, Romania; dr.razvanp@gmail.com; 3Urology Department, Carol Davila University of Medicine and Pharmacy, 020021 Bucharest, Romania; 4Othorhinolaryngology and Ophtalmology Department, Carol Davila University of Medicine and Pharmacy, 020021 Bucharest, Romania; andreearusescu@gmail.com

**Keywords:** renal abscess, SGLT2 inhibitors, heart failure, urinary tract infections

## Abstract

**Background and Clinical Significance:** Renal abscess represents one infectious urological complication with lethal potential. The treatment of this pathology may differ depending on the severity of the symptoms and the size of the infectious collection. Diabetes, immunosuppression, and associated urinary pathologies are most frequently responsible for the development of abscesses. This case report presents the first documented case of a renal abscess associated with Sodium-glucose cotransporter 2 (SGLT2) inhibitors in a person without previous predisposing pathologies. **Case Presentation:** A 62-year-old patient presented to the emergency department for pain in the right flank, vomiting, and dysuria. The patient’s medical history revealed Heart Failure New York Heart Association (NYHA) Class II, Coronary Artery Disease (CAD) with prior angioplasty, and permanent Atrial Fibrillation. No prior urological or immunosuppressive conditions were detected. The Computed Tomography (CT) evaluation confirmed the ultrasound suspicion of a right renal abscess performed in the emergency room. The only risk factor identified was the initiation of SGLT2 inhibitor therapy for cardiac pathology approximately 2 months before. According to the small dimensions and urine culture, the abscess was successfully treated with antibiotic administration in collaboration with the urology department. The infectious process was remitted within 2 weeks. **Conclusions:** To our knowledge this is the first documented case of a renal abscess associated with SGLT2 inhibitor administration in a person without previous predisposing risk factors. Despite the relatively low incidence of urinary tract infections (UTIs) associated with SGLT2 inhibitors, their widespread use in the treatment of various socially significant conditions highlights the need for both patients and medical specialists to be aware of all potential risks and pay increased attention to these cases.

## 1. Introduction and Clinical Significance

Sodium-glucose cotransporter 2 (SGLT2) inhibitors have become a significant therapeutic option for the management of heart failure (HF), demonstrating efficacy that extends beyond their initial function in glycemic control for patients with type 2 diabetes mellitus (T2DM). Empagliflozin and dapagliflozin, some of the most used SGLT2 inhibitors, have been shown to substantially reduce the risk of hospitalization and mortality associated with HF, regardless of the presence of diabetes, according to clinical evidence. Inhibiting SGLT2 reduces the renal threshold for glucose, disrupting tubular glucose reabsorption [1,2].

Glucose facilitates the excretion of free water through osmotic diuresis, and the surplus glucose retained in the tubule is eliminated in the urine, resulting in glycosuria (70–80 g/day), which increases the patient’s susceptibility to bacterial infection [3]. Despite the absence of anticipated complications, a list of severe adverse events restricts the prescription of SGLT2 inhibitors in certain patients [2,3]. Genitourinary infections are documented side effects that may arise following the treatment of SGLT2 inhibitors [3].

Significant urinary tract infections (UTIs), including urosepsis and pyelonephritis, along with numerous genital infections such as genital warts, mycotic infections, urethritis, and balanitis, have also been documented [3,4,5,6]. Although SGLT2 inhibitors may induce glycosuria, elevate the risk of UTIs, and foster an environment conducive to bacterial proliferation, certain studies indicate that the risk of developing urosepsis is comparable to that associated with other antidiabetic agents. A plausible explanation for this phenomenon is attributed to higher urine output resulting from osmotic diuresis and natriuresis [4,5,6].

Prior meta-analyses have reported inconsistent results regarding the link between SGLT2 inhibitors and UTIs, even though they have consistently been proven to increase the incidence of genital infections [3,4,5,6]. The US Food and Drug Administration (FDA) updated the labels for all SGLT2 inhibitors in 2015, adding a warning for severe UTIs, even though the majority of UTIs brought on by these medications were mild to moderate in intensity. Post-marketing adverse event reports of pyelonephritis and sepsis with UTIs in individuals using these medications prompted this warning [4,5,6].

Renal abscesses represent one of the infectious urological complications with lethal potential. The treatment of this pathology may differ depending on the severity of the symptoms and the size of the infectious collection. Diabetes, immunosuppression, and associated urinary pathologies are most frequently responsible for the development of abscesses. These may result from an ascending infection or bacterial dissemination from another septic source within the body [7,8,9]. Specific risk factors, including structural anomalies of the genitourinary tract, a history of complex or recurrent UTI, nephrolithiasis and blockage, immunodeficiency, pregnancy, and diabetes, may predispose individuals to the development of renal abscesses [7,8,9]. Early diagnosis is essential to avoid additional complications and decrease morbidity.

Diagnosing a renal abscess is challenging due to its frequent presentation with nonspecific symptoms, including fever, chills, and abdomen or flank pain. Prior intricate instances of renal abscess have been documented in [7,9,10,11,12,13], although most of these patients possess at least one risk factor.

The functions of the kidney and the heart are mutually dependent. The kidney disease or malfunction may cause, worsen, or trigger the heart disease or dysfunction, and vice versa, creating a vicious cycle. Furthermore, while the kidney and heart have a reciprocal relationship, additional illnesses, including diabetes, hypertension, and atherosclerosis, may coexist and affect both organs at the same time [13,14].

To our knowledge this is the first documented case of a renal abscess associated with SGLT2 inhibitors administration in a person without previous predisposing risk factors. Despite the relatively low incidence of urinary tract infections associated with SGLT2 inhibitors, this study describes the case of renal abscess diagnosed in a patient treated with Dapagliflozin for HF with reduced ejection fraction with no other previous risk factors.

## 2. Case Presentation

A 62-year-old Caucasian male presented to the emergency department (ED) with right flank pain radiating to the groin, dizziness, vomiting, and lightheadedness for 3 days, with the presence of urinary symptoms like dysuria without fever. His medical history was relevant for HF NYHA Class II, Coronary Artery Disease (CAD) with prior angioplasty and permanent Atrial Fibrillation. Dapagliflozin 10 mg per day had been introduced 2 months before this presentation. The patient underwent a thorough urology evaluation 3 months before Dapagliflozin initiation. Key findings included a prostate-specific antigen (PSA) level of just 1 ng/mL and a negative urine culture, indicating no signs of infection. A digital rectal examination confirmed a flat and painless prostate, further reassuring the clinical case. Additionally, a renal ultrasound demonstrated normal kidney function, featuring non-dilated collecting systems and a complete absence of urinary stones. Also, a prostatic volume of 25 cm and no residual voiding urine were observed. This comprehensive assessment confirms the patient’s suitability for Dapagliflozin therapy, maximizing the potential for positive outcomes in HF with a reduced ejection fraction. The patient’s atrial fibrillation was managed with Apixaban 5 mg twice a day and CAD with Clopidogrel 75 mg/day. His hypertension was managed with Perindopril 5 mg daily, and his dyslipidemia with Atorvastatin 80 mg/day. The patient was a former smoker and had ceased smoking 2 years ago. The patient’s family history revealed heart disease, hypertension, diabetes, and dyslipidemia. Upon physical examination, the patient’s body mass index was recorded as 32 kg/m^2^. Initial vital signs showed a T-max of 36.8 °C, a heart rate (HR) of 97 beats per minute, a blood pressure of 136/86 mm Hg, and a respiratory rate of 17 breaths per minute. Examination of other systems was nonrevealing. ECG showed atrial fibrillation with HR = 97/min. Transthoracic echocardiography (TTE) showed left ventricular hypertrophy (LVH) with impaired systolic function [left ventricular ejection fraction (LVEF) = 39%]; hipokinesia in the basal third of the interventricular septum (IVS); left atrial enlargement; restrictive LV filling patterns; LV mass = 142 g/m^2^; stroke volume (SV) = 42 mL/beat; indexed SV = 20 mL/beat/m^2^; mitral annular plane systolic excursion (MAPSE) = 8 mm; a moderate ischemic and degenerative mitral regurgitation; a moderate tricuspid regurgitation; mild secondary pulmonary hypertension (PASP = 43 mmHg); and a global longitudinal strain of 7.5%. (Table 1). Also, a 2D echocardiography revealed no vegetation.

### 2.1. Diagnostic Assessment

Laboratory evaluation showed leukocytosis, and a high level of C reactive protein at baseline. The detailed results of his blood count and other pertinent labs are shown in Table 2. The Urine blood culture was positive for a UTI with Escherichia coli. The *E. coli* was sensitive to amikacin, cefoperazone and/or sulbactam, gentamicin, imipenem, meropenem, natamycin, nitrofurantoin, and piperacillin and/or tazobactam. An enhanced CT scan was performed based on abdominal US suspicion of a renal mass. The imagistic evaluation showed fluid mass in the inferior pole of the right kidney with surrounding inflammatory changes concerning renal abscess, measuring 15.3 × 21.8 mm, as shown in Figure 1. A differential diagnosis was made for an infected renal cyst. There were inflammatory changes in the right renal collecting system associated with the thickening of the urothelium and bladder wall without any obstructive pathologies or secondary determining hydronephrosis.

### 2.2. Management

Based on the radiology imagistic interpretation of the mass as a renal abscess, broad-spectrum antibiotic treatment (Meropenem 3 g/daily) was started according to the urology and infectious disease departments’ recommendations.

Active fluid and electrolyte supplementation and symptomatic medications were administered. According to the active guideline recommendations for small renal abscesses (<3 cm), no percutaneous or open drainage were necessary as the initial treatment in this case [14]. The patient’s condition was periodically monitored by blood count assessment and ultrasonographic evaluation during his hospital stay. The follow-up enhanced CT scan performed on the 14th day of hospitalization revealed that the renal collection was in complete remission (Figure 2). He was discharged in stable condition after a few days, and Dapagliflozin was withheld from the patient.

## 3. Discussion

Six oral SGLT2 inhibitors are currently approved for the treatment of T2DM by the FDA together with the European Medicines Agency (EMA): canagliflozin (CANA), empagliflozin (EMPA), dapagliflozin (DAPA), ertugliflozin (ERTU), bexagliflozin, and sotagliflozin [15]. SGLT2 inhibitors are extensively recommended owing to their advantageous impact on renal and cardiovascular outcomes in both diabetic and nondiabetic patients. Also, SGLT2 inhibitors represent the most recent advancement in HF treatment and are the sole pharmacological class demonstrating efficacy across the whole spectrum of HF, irrespective of LVEF [16]. The Multicenter Trial to Evaluate the Effect of Dapagliflozin on the Incidence of Cardiovascular Events (DECLARE-TIMI58) trial demonstrated no significant change in the occurrence of major adverse cardiovascular events among patients administered Dapagliflozin; however, it indicated a reduction in cardiovascular mortality and hospitalizations associated with HF [17].

Additionally, the beneficial effects of these medications on cardiovascular health are believed to be achieved through the promotion of weight loss, a reduction in blood pressure, a reduction in vascular resistance, and a reduction in serum uric acid levels [17,18,19,20,21]. The advantages are associated with renovascular protection resulting from reduced hyperfiltration, triggered by elevated sodium concentrations activating tubuloglomerular feedback in the macula densa, which leads to afferent vasomodulation [21,22].

A recent meta-analysis of 13 trials demonstrated a 37% reduction in the relative risk of renal disease progression, a 23% reduction in the risk of acute kidney injury, and a 23% reduction in the risk of cardiovascular death or hospitalization for HF with comparable effects observed in patients both with and without diabetes [19,20,21,22,23]. The consequences that relate to SGLT2 inhibitors were also brought to light in this study. These issues included the risk of developing a UTI (relative risk 1.08, 95% confidence interval 1.02–1.15), as well as mycotic genital infections (relative risk 3.57, 95% confidence interval 3.14–4.06) [23].

Currently, there are limited case reports documenting patients undergoing SGLT2 inhibitor therapy who have developed renal abscesses. All reported cases involved patients with comorbidities, including diabetes mellitus, congenital or acquired obstructive uropathies, or those who underwent immunosuppressive treatment [19,20,21,22,23].

In just 14 patients in the three landmark investigations on SGLT2 and cardiovascular outcomes [17,24,25], these drugs were linked to a higher risk of genital infections. The report suggests that elevated doses of dapagliflozin could correlate with an increased risk of UTI [26].

The most prevalent species responsible for renal abscesses are Escherichia coli, which was present in our patient, Proteus, and Staphylococcus aureus [26].

On a quick search on PubMed and Google, there were only three cases of renal abscess linked to SGLT2 inhibitors, but in all cases, there were also some other predisposing pathologies. All the identified cases are illustrated in Table 3.

In contrast to previous reports, this case presents an adult male patient without any underlying pathologies like diabetes mellitus, immunosuppression, or obstructed drainage, conditions commonly associated with renal abscesses [27,28,29]. The sole potential predisposing factor is the administration of an SGLT2 inhibitor. Even though many reports indicate that fungal infections related to SGLT2 administration are responsible for different kinds of abscess, the only urinary tract bacteria identified were *E. coli* [28]. Another important factor illustrated by the present case is that prompt diagnosis and antibiotic treatment may prevent further complications requiring surgical management [27,28,29].

In 1996, Siegel presented a therapy regimen for renal abscesses based on their dimensions. According to its dimensions (<3 cm), this abscess was classified as minor, and broad-spectrum intravenous antibiotic therapy was administered. In contrast, invasive management, such as percutaneous drainage or open surgery, was advised for large renal abscesses (more than 3 or 5 cm) [10]. The decision between a conservative or operative approach for this case was also based on the response to antimicrobial therapy and the patient’s clinical status.

Educating patients on recognizing the early signs of a UTI and encouraging them to report them can help to minimize the risk of complications. Although the use of SGLT2 inhibitors has not been linked to UTIs, clinicians should be cautious when prescribing these drugs to patients with serious or recurrent urogenital infections, abnormal urinary flow (e.g., incomplete bladder emptying with urinary stasis), or indwelling Foley catheters.

Phytotherapy has been explored for preventing UTIs, and several herbal agents have shown potential in reducing UTI recurrence, like cranberry (*Vaccinum macrocarpon*), D-mannose, and herbal blends [30]. Nutraceuticals or phytotherapy may be a new way to manage recurrent UTI patients after assessing their risk factors. There is no optimum ingredient or mixture for preventing recurrent UTIs.

Plant-based antibiotic alternatives are cost-effective, readily available, safe, have fewer side effects, reduce antimicrobial resistance risks, and reduce the adverse effects and symptoms of antibiotics for the prevention and treatment of recurrent UTIs [31]. A tailored approach according to bacterial characteristics and the patient’s risk factor profile is a promising option.

## 4. Conclusions

Although the reported rate of urinary infections associated with the administration of SGLT2 inhibitors is relatively low, unspecific manifestations like renal abscesses may occur even in patients with no previous risk factors.

According to the actual guidelines, it is not recommended to treat asymptomatic bacteriuria because of the increasing bacterial resistance rates, but it might be useful for both patients and clinicians to be aware of when starting SGLT2 inhibitors and it is important for patients to present themselves to a hospital unit from the early onset of symptoms linked to urinary tract infections.

## Figures and Tables

**Figure 1 biomedicines-13-00389-f001:**
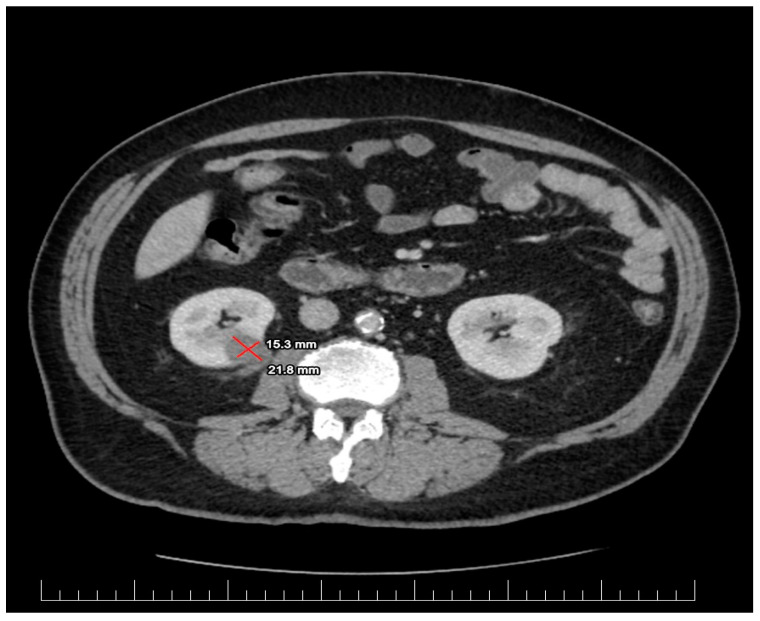
The first CT scan evaluation revealing lower pole fluid collection measuring 15.3/21.8 mm suggestive for renal abscess.

**Figure 2 biomedicines-13-00389-f002:**
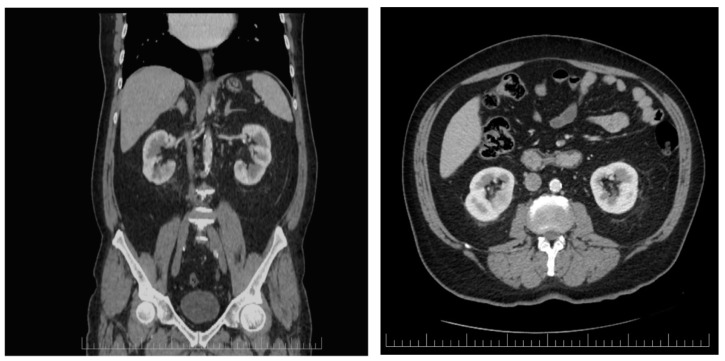
Axial and coronal enhaced CT scan performed after 2 weeks of continuous antibiotic administration revealing that the renal abscess completely dissapeared.

**Table 1 biomedicines-13-00389-t001:** Baseline echocardiographic parameters. LV—left ventricle; EDD—end diastolic dimension; ESD—end systolic dimension; EDV—end diastolic volume; ESV—end systolic volume; EF—ejection fraction; MAPSE—mitral annular plane systolic excursion; PASP—pulmonary artery systolic pressure.

LV-EDD (mm)	60
LV-ESD (mm)	50
LV-EDV (mL)	151
LV-ESV (mL)	109
LV-EF (%)	39
LV mass index (g/m^2^)	142
LA volume index (mL/m^2^)	60
Mitral annular e′ velocity (cm/s)	4.7
Mitral annular s′ velocity (cm/s)	4.7
Mitral annular E/e′ ratio	21
MAPSE (mm)	8
PASP (mmHg)	43
Global longitudinal strain (%)	7.5

**Table 2 biomedicines-13-00389-t002:** The results of routine laboratory tests performed at admission.

Lab	Results	Normal Range
WBC	**12,100/mm^3^ (25.1 × 10^9^/L)**	4500–10,000/mm^3^ (4.5–11.0 × 10^9^/L)
Bands	**21% (21)**	0–3% (0–3)
Hemoglobin	**15.3 g/dL (6.95 mmol/L)**	13.5–17.5 g/dL (8.38–10.86 mmol/L)
Hematocrit	**38.3% (38)**	39–49% (39–49)
Platelet count	144,000/mm^3^ (144 × 10^9^/L)	150,000–400,000/mm^3^ (150–400 × 10^9^/L)
BUN	**34 mg/dL (10.7 mmol/L)**	8–25 mg/dL (2.9–8.9 mmol/L)
Creatinine	**1.21 mg/dL (314.71 µmol/L)**	0.5–1 mg/dL (74.3–107 µmol/L)
Sodium	**140 mmol/L**	135–145 mmol/L (135–145 mmol/L)
Potassium	5.4 mmol/L	3.4–5.0 mmol/L (3.4–5.0 mmol/L)
Bicarbonate	**21 mmol/L**	20–32 mmol/L (20–32 mmol/L)
Procalcitonin	**0.03 ng/mL (0.03 µg/L)**	0.05 ng/mL (0.05 µg/L)
Lactic acid	**2.1 mmol (2.1 mmol/L)**	0.5–2.2 mmol/L (0.5–2.2 mmol/L)
Hemoglobin A1C	**5.4%**	<6%
Blood glucose C reactive protein	**110 mg/dL (16.04 mmol/L)** **20 mg/L**	70–110 mg/dL (3.9–6.1 mmol/L) 0.10–5.00 mg/L

Abbreviations: BUN—blood urea nitrogen; WBC—white blood cells. Abnormal values are shown in bold font. Values in parenthesis are the international system of units.

**Table 3 biomedicines-13-00389-t003:** Cases of renal abscess linked to SGLT2 inhibitors.

Nr	Study Details	Underlying Disease	Bacterial Agent	Renal Manifestation	Treatment	SGLT2 Inhibitor
1.	(Pablo Echeveria et al., 2023) [27]	T2DM Hypertension Dyslipidemia	*E. coli*	Emphysematous Pyelonephritis	Nephrectomy	Empaglifozin
2.	(Prathap Kumar Simhadri et al., 2024) [28]	T2DM Hypertension CKD BPH	*Candida tropicalis*	Renal Abscess	Percutaneous drainage	Canaglifozin
3.	(Kazuki Yanagida et al., 2024) [29]	T2DM	Not Specified	Renal Abscess	Percutaneous drainage Open surgical drainage and HBO_2_ therapy	Not specified

Abbreviations: CKD—chronic kidney disease; BPH—benign prostate enlargement.

## Data Availability

Data are contained within the article.

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
