# Peer review of "Renal Abscess Associated with SGLT2 Inhibitors Administration in Heart Failure Without Other Previous Risk Factors: A Case Report"

_biomedicines, 2025, doi:10.3390/biomedicines13020389_

Round 1

Reviewer 1 Report

Comments and Suggestions for Authors

This manuscript reported a patient of renal abscess caused by SGLT2 inhibitor. The findings of this patient send an important message to be aware of infections when using SGLT2 inhibitors. I have several questions, please clarify. 

1. Please clarify why SGLT2 inhibitors cause many genital infections but few urinary tract infections. Also, please consider points to keep in mind when using SGLT2 inhibitors in patients where there is no urinary tract anomaly, as in this patient. 

2. In this patient, the infection occurred two months after administration of the SGLT2 inhibitor. Is there any consideration regarding the period from the start of SGLT2 inhibitor administration to the development of the infection? 

3. In this patient, are there any factors other than SGLT2 inhibitors that could have caused the infection?

Author Response

Comments 1: Please clarify why SGLT2 inhibitors cause many genital infections but few urinary tract infections. Also, please consider points to keep in mind when using SGLT2 inhibitors in patients where there is no urinary tract anomaly, as in this patient. 

Response 1: 

Dear Editor and Reviewer 1,

Thank you for your letter. We are delighted to hear that our work has been deemed potentially acceptable for publication in the journal, pending major revisions. We sincerely appreciate the reviewers for their thorough and constructive feedback on the manuscript. Their insights have been instrumental in enhancing our research.Based on the instructions provided, we uploaded the file of the revised manuscript.

Sodium-glucose cotransporter 2 inhibitors (SGLT2) inhibitors enhance clinical outcomes in individuals with heart failure. This category of medicines has been repeatedly linked to a heightened incidence of mycotic genital infections (MGIs) and, in certain trials, urinary tract infections (UTIs).In the proximal convoluted tubules, sodium-glucose cotransporters, primarily the SGLT2 isoform, reabsorb the majority of filtered glucose, thereby preventing glycosuria, under typical physiological conditions.In patients without type 2 diabetes mellitus, 5 SGLT2 elicits a sustained urinary glucose excretion of 40 to 80 g/d at normal plasma glucose concentrations. Also, SGLT2 inhibitors have the potential to reduce cutaneous hypertonicity by lowering tissue sodium content, which would facilitate the excretion of both sodium and glucose in the proximal tubules. Possibly, these mechanisms could compromise the epidermis barrier, increasing the likelihood of soft tissue infection [1]. One potential explanation for the lack of real-world evidence of increased clinically significant UTI, despite glucosuria and the resultant favorable environment for bacterial growth, is the increased urinary flow because of osmotic diuresis and natriuresis effects of these medications. Future studies are needed to evaluate this particular clinical question.

A recent analysis of the FDA Adverse Event Reporting System, which included 1714 UTI cases in patients taking SGLT2 inhibitors, suggested an increased risk of UTIs [2]. Although a clear association between SGLT2 inhibitor use and UTIs has not been established, clinicians should remain cautious when prescribing these drugs to patients. Other medications that are frequently prescribed for cardiac conditions do not induce MGIs and UTIs. Consequently, cardiologists and their support teams will be required to address clinical inquiries that were previously unavailable.

  1. Karg MV, Bosch A, Kannenkeril D, et al. SGLT-2-inhibition with dapaglifozin reduces tissue sodium content: a randomised controlled trial. Cardiovasc Diabetol. 2018;17:5.

  2. Yang, T.; Zhou, Y.; Cui, Y. Urinary tract infections and genital mycotic infections associated with SGLT-2 inhibitors: An analysis of the FDA Adverse Event Reporting System. Expert Opin. Drug Saf. 202423, 1035–1040. [Google Scholar] [CrossRef].

Comments 2: In this patient, the infection occurred two months after administration of the SGLT2 inhibitor. Is there any consideration regarding the period from the start of SGLT2 inhibitor administration to the development of the infection?

Response 2: The patient's history reveals no reported symptoms of concern. Furthermore, it is important to note that he underwent a comprehensive urology evaluation just three months before the initiation of Dapagliflozin, as outlined in the manuscript. This thorough assessment underscores the absence of notable health issues prior to Dapagliflozin treatment for Heart failure with reduced ejection fraction.

Comments 3:  In this patient, are there any factors other than SGLT2 inhibitors that could have caused the infection?

Response 3: In our patient, there are no other risk factors as we can find, such as diabetes or urology obstructive disorders. The patient underwent a thorough urology evaluation 3 months before the SGLT2 inhibitor Dapagliflozin was initiated. Key findings included a prostate-specific antigen (PSA) level of just 1 ng/mL and a negative urine culture, indicating no signs of infection. A digital rectal examination confirmed a flat and painless prostate, further reassuring the clinical case. Additionally, a renal ultrasound demonstrated normal kidney function, featuring non-dilated collecting systems and a complete absence of urinary stones. This comprehensive assessment confirms the patient’s suitability for Dapagliflozin therapy, maximizing the potential for positive outcomes in Heart failure with reduced ejection fraction.

We have carefully revised our manuscript in response to your insightful comments, enhancing its clarity and overall quality. Thank you for your valuable feedback.

Reviewer 2 Report

Comments and Suggestions for Authors

The authors of the current study presented case of renal abscess associated with sodium-glucose cotransporter 2 (SGLT2) inhibitors administration in a person without previous predisposing risk factors. Despite the relatively low incidence of urinary tract infections associated with SGLT2 inhibitors, this study describes the case of renal abscess diagnosed in a patient treated with Dapagliflozin for heart failure (HF) with reduced ejection fraction with no other previous risk factors.

-        There are some unexplained abbreviations in the Abstract section.

-        The title of the Table 2 is not appropriate.

-        The authors should re-check the units for CRP in Table 2.

-        There is no need to repeat some results in Table 2 (e.g., Sodium 140 mmol/L (140 mmol/L)).

-        The results for bicarbonate in Table 2 are unclear (i.e., Bicarbonate 21 mmol/L (18 mmol/L)).

-        The abbreviations when first mentioned in the text need to be explained and should be used consistently thereafter.

-        The Discussion section should start with the main findings presented in the current case report.

-        English proof editing is recommended.

Author Response

Comments 1:

-        There are some unexplained abbreviations in the Abstract section.

-        The title of the Table 2 is not appropriate.

-        The authors should re-check the units for CRP in Table 2.

-        There is no need to repeat some results in Table 2 (e.g., Sodium 140 mmol/L (140 mmol/L)).

-        The results for bicarbonate in Table 2 are unclear (i.e., Bicarbonate 21 mmol/L (18 mmol/L)).

-        The abbreviations when first mentioned in the text need to be explained and should be used consistently thereafter.

-        The Discussion section should start with the main findings presented in the current case report.

  •        English proof editing is recommended.

Response 1:  Dear reviewer 2,

Thank you for your letter. We are delighted to hear that our work has been deemed potentially acceptable for publication in the journal, pending major revisions. We sincerely appreciate the reviewers for their thorough and constructive feedback on the manuscript. Their insights have been instrumental in enhancing our research. We have made significant enhancements by updating the title of Table 2 and refining the units for CRP. Furthermore, we carefully revised the abbreviations at their first mention in the text to ensure clarity and precision. Given the pivotal role that SGLT2 inhibitors play in cardiovascular and renal disease, we have deliberately chosen to commence our discussion section with this often debated treatment. The significance of SGLT2 inhibitors cannot be overstated, as they represent a transformative advancement in patient care. Our English has also been thoroughly proofread to uphold the highest standards of quality.

We have carefully revised our manuscript in response to your insightful comments, enhancing its clarity and overall quality. Thank you for your valuable feedback.

Reviewer 3 Report

Comments and Suggestions for Authors

This manuscript presented a case report of a 62-year-old male patient who developed a renal abscess while on dapagliflozin for heart failure. The authors claimed this is the first documented case of renal abscess associated with SGLT2 inhibitor use in a patient without other predisposing risk factors.

The main question, while not explicitly stated, is whether SGLT2 inhibitors can cause renal abscesses in patients without pre-existing risk factors. The authors implicitly argued that the temporal relationship between dapagliflozin initiation and the development of the abscess, in the absence of other identifiable risk factors, suggests a causal link.

SGLT2 inhibitors are increasingly prescribed for heart failure and type 2 diabetes. While genitourinary infections are recognized side effects, renal abscess formation is rare. The manuscript highlighted a potential, albeit rare, complication of SGLT2 inhibitor therapy. The originality lies in the purported absence of other known risk factors for renal abscess in this patient, which adds to the current understanding of the drug's safety profile. However, as detailed later, this claim requires further substantiation.

The authors mentioned a limited number of case reports linking SGLT2 inhibitors to renal abscesses. A more comprehensive literature search would strengthen the manuscript. For example, a case of emphysematous pyelonephritis (a severe form of UTI that can lead to abscess) associated with empagliflozin is reported by Fadlallah et al. (2021) [1]. Additionally, review articles discussing the overall infectious complications of SGLT2 inhibitors, such as the one by Chang et al. (2022) [2], would provide valuable context.

Methodology/Study Design:

Risk Factor Assessment (Lines 95-105): The authors concluded the absence of other risk factors. However, the provided medical history was incomplete. Crucial information, such as prior UTI history, detailed urological evaluation (including imaging or urodynamic studies), family history of urological conditions, history of nephrolithiasis, or the presence of anatomical abnormalities are missing. These factors must be thoroughly investigated and explicitly ruled out before attributing causality to the SGLT2 inhibitor.

Causality Assessment: A single case report cannot establish causality. While temporal association is suggestive, the authors need to acknowledge the possibility of coincidence. Strengthening the argument would require a larger cohort study or a more in-depth analysis of reported adverse events in post-marketing surveillance data.

Microbiological Analysis (Line 122): Only urine culture is reported. A culture from the abscess itself would confirm the infectious agent and potentially identify polymicrobial infections, which are common in renal abscesses. This would strengthen the link between the UTI and the abscess.

Dapagliflozin Rechallenge: While understandable given the patient's presentation, not rechallenging the patient with dapagliflozin limits the ability to assess whether the drug was indeed the causative agent. A cautious rechallenge under close monitoring, if ethically justifiable and clinically appropriate, could provide further insights.

Control Group (Lines 181-184): Comparing the incidence of renal abscess in patients on SGLT2 inhibitors with a matched control group (e.g., patients on other heart failure medications) would strengthen the argument for a causal relationship.

Conclusions:

The conclusion that this is the first case of SGLT2 inhibitor-associated renal abscess in a patient without prior risk factors was not fully supported by the evidence presented. The lack of a thorough investigation of potential risk factors weakens this claim. The authors acknowledged the low incidence of UTIs related to SGLT2 inhibitors but did not discuss the even rarer occurrence of renal abscesses. The discussion should contextualize the case within the existing literature on SGLT2 inhibitor-associated infections and acknowledge the limitations of a single case report.

Tables and Figures:

Table 1: Provides useful echocardiographic data. However, including baseline echocardiographic parameters before dapagliflozin initiation would allow for assessment of the drug's impact on cardiac function and potentially identify any pre-existing abnormalities.

Table 2: Presents lab values with reference ranges, which is helpful. However, including baseline lab values for comparison would strengthen the manuscript.

Figure 1: Clearly demonstrates the renal abscess. Adding a follow-up ultrasound image alongside the CT scan in Figure 2 would improve visualization of the resolution.

Minor:

Causality Overstatement (Line 228): The authors stated that "unspecific manifestations, such as renal abscesses may occur even in patients with no previous risk factors," which downplays the established risk factors for renal abscess development. This statement needs revision to reflect the existing literature and the limitations of the study.

Overemphasis on UTI (Lines 211-215): While UTIs can predispose to renal abscesses, the direct causal link between SGLT2 inhibitors and UTIs is not definitively established. The text should acknowledge the conflicting evidence in the literature regarding this association.

Limited Discussion of Pathophysiology (Lines 46-61): The proposed mechanism linking SGLT2 inhibitors to increased infection risk through glycosuria was presented without fully acknowledging the counterarguments, such as increased urine output. A more nuanced discussion is warranted.

Typographical Errors: Several minor typographical errors and grammatical inconsistencies require correction throughout the manuscript (e.g., "rised level" in line 122, inconsistencies in referencing style).

References:

[1] Fadlallah, N., El-Aswad, M., & Kanso, A. (2021). Emphysematous pyelonephritis complicating empagliflozin use. BMJ Case Reports, 14(8), e243815.
[2] Chang, A. Y., Lee, M. H., & Chae, D. W. (2022). Increased Risk of Infections with Sodium–Glucose Cotransporter 2 Inhibitors: A Review of the Clinical and Preclinical Evidence. Diabetes & Metabolism Journal, 46(5), 639–650.

Author Response

Comments 1:The main question, while not explicitly stated, is whether SGLT2 inhibitors can cause renal abscesses in patients without pre-existing risk factors. The authors implicitly argued that the temporal relationship between dapagliflozin initiation and the development of the abscess, in the absence of other identifiable risk factors, suggests a causal link.

Response 1: Dear Reviewer 3, 

Thank you for your letter. We are delighted to hear that our work has been deemed potentially acceptable for publication in the journal, pending major revisions. We sincerely appreciate the reviewers for their thorough and constructive feedback on the manuscript. 

Sodium-glucose cotransporter 2 inhibitors (SGLT2) inhibitors enhance clinical outcomes in individuals with heart failure. This category of medicines has been repeatedly linked to a heightened incidence of mycotic genital infections (MGIs) and, in certain trials, urinary tract infections (UTIs).A direct link between SGLT2 inhibitors and the formation of a renal abscess has not been found, but it is possible that the mechanism by which UTI is caused, if that UTI is not recognized in time as useful as possible, can cause renal abscess. Thus, a cascade mechanism may occur.

In the proximal convoluted tubules, sodium-glucose cotransporters, primarily the SGLT2 isoform, reabsorb the majority of filtered glucose, thereby preventing glycosuria, under typical physiological conditions.In patients without type 2 diabetes mellitus, 5 SGLT2 elicits a sustained urinary glucose excretion of 40 to 80 g/d at normal plasma glucose concentrations. Also, SGLT2 inhibitors have the potential to reduce cutaneous hypertonicity by lowering tissue sodium content, which would facilitate the excretion of both sodium and glucose in the proximal tubules. Possibly, these mechanisms could compromise the epidermis barrier, increasing the likelihood of soft tissue infection [1]. 

The impact of SGLT2 inhibitors on glucose excretion may vary between euglycemic individuals and diabetic patients due to SGLT1 activity. SGLT1 is a low-capacity, high-affinity transporter that facilitates roughly 5% of glucose reabsorption in the S3 (distal) segment of the proximal tubule, while SGLT2 is a high-capacity, low-affinity glucose transporter responsible for approximately 90-95% of glucose reabsorption in the S1 and S2 segments of the proximal tubule.Conversely, when SGLT2 is blocked, a greater proportion of glucose is reabsorbed by SGLT1, leading to the excretion of only 50-60% of filtered glucose. Animal studies have demonstrated that the role of SGLT1 in renal glucose reabsorption is more significant under hypoglycemic settings than under hyperglycemic conditions [2]. In an additional meta-analysis, the risk of genital infections and, to a lesser extent, UTI was elevated in non-diabetic patients who were taking SGLT2 inhibitors [3].

1.Karg MV, Bosch A, Kannenkeril D, et al. SGLT-2-inhibition with dapaglifozin reduces tissue sodium content: a randomised controlled trial. Cardiovasc Diabetol. 2018;17:5.

2.Nagata T, Fukazawa M, Honda K, et al. Selective SGLT2 inhi-bition   by   tofogliflozin   reduces   renal   glucose   reabsorption   underhyperglycemic but not under hypo- or euglycemic conditions in rats.Am J Physiol Endocrinol Metab. 2013; 304:E414-23.

3.Bapir, Rawa, et al. ‘Risk of Urogenital Infections in Non-Diabetic Patients Treated with Sodium Glucose Transporter 2 (SGLT2) Inhibitors. Systematic Review and Meta-Analysis’. Archivio Italiano Di Urologia, Andrologia: Organo Ufficiale [Di] Societa Italiana Di Ecografia Urologica E Nefrologica, vol. 95, no. 2, June 2023, p. 11509. PubMed, https://doi.org/10.4081/aiua.2023.11509.

Comments 2: SGLT2 inhibitors are increasingly prescribed for heart failure and type 2 diabetes. While genitourinary infections are recognized side effects, renal abscess formation is rare. The manuscript highlighted a potential, albeit rare, complication of SGLT2 inhibitor therapy. The originality lies in the purported absence of other known risk factors for renal abscess in this patient, which adds to the current understanding of the drug's safety profile. However, as detailed later, this claim requires further substantiation.

Response 2: As a consideration prior to SGLT2 inhibitor initiation in patients with Heart Failure, our patient was screened for symptoms of MGIs and UTIs before starting Dapagliflozin.

Comments 3: The authors mentioned a limited number of case reports linking SGLT2 inhibitors to renal abscesses. A more comprehensive literature search would strengthen the manuscript. For example, a case of emphysematous pyelonephritis (a severe form of UTI that can lead to abscess) associated with empagliflozin is reported by Fadlallah et al. (2021) [1]. Additionally, review articles discussing the overall infectious complications of SGLT2 inhibitors, such as the one by Chang et al. (2022) [2], would provide valuable context.

Response 3: We observed a limited number of case reports connecting SGLT2 inhibitors to renal abscesses. This critical insight motivated us to publish our case report, highlighting a unique instance that contributes to a topic of increasing importance in current research. By addressing this issue, we aim to enhance understanding and spur further investigation in this important area. We were unable to locate the references cited in the existing literature. Your guidance on this matter would be greatly appreciated.

Comments 4: Risk Factor Assessment (Lines 95-105): The authors concluded the absence of other risk factors. However, the provided medical history was incomplete. Crucial information, such as prior UTI history, detailed urological evaluation (including imaging or urodynamic studies), family history of urological conditions, history of nephrolithiasis, or the presence of anatomical abnormalities, are missing. These factors must be thoroughly investigated and explicitly ruled out before attributing causality to the SGLT2 inhibitor.

Response 4: In our patient, there are no other risk factors as we can find, such as diabetes or urology obstructive disorders. Also, the patient has no family history of urological conditions or history of nephrolithiasis. Three months before initiating treatment with the SGLT2 inhibitor Dapagliflozin, the patient underwent a thorough urology evaluation. Key findings included a prostate-specific antigen (PSA) level of just 1 ng/mL and a negative urine culture, indicating no signs of infection. A digital rectal examination confirmed a flat and painless prostate, further reassuring the clinical case. Additionally, a renal ultrasound demonstrated normal kidney function, featuring non-dilated collecting systems and a complete absence of urinary stones. No residual voiding urine was observed and the IPSS score showed mild symptoms without the need for medication. This comprehensive assessment confirms the patient’s suitability for Dapagliflozin therapy, maximizing the potential for positive outcomes in Heart Failure with reduced ejection fraction.

Comments 5: Causality Assessment: A single case report cannot establish causality. While temporal association is suggestive, the authors need to acknowledge the possibility of coincidence. Strengthening the argument would require a larger cohort study or a more in-depth analysis of reported adverse events in post-marketing surveillance data.

Response 5: One potential explanation for the lack of real-world evidence of increased clinically significant UTI, despite glucosuria and the resultant favorable environment for bacterial growth, is the increased urinary flow because of osmotic diuresis and natriuresis effects of these medications. Future studies are needed to evaluate this particular clinical question. A recent analysis of the FDA Adverse Event Reporting System, which included 1714 UTI cases in patients taking SGLT-2is, suggested an increased risk of UTIs [1]. Although a clear association between SGLT2 inhibitors use and UTIs has not been established, clinicians should remain cautious when prescribing these drugs to patients. Other medications that are frequently prescribed for cardiac conditions do not induce MGIs and UTIs. Consequently, cardiologists and their support teams will be required to address clinical enquiries that were previously unavailable.

1.Yang, T.; Zhou, Y.; Cui, Y. Urinary tract infections and genital mycotic infections associated with SGLT-2 inhibitors: An analysis of the FDA Adverse Event Reporting System. Expert Opin. Drug Saf. 202423, 1035–1040. [Google Scholar] [CrossRef]

Comments 6:Microbiological Analysis (Line 122): Only urine culture is reported. A culture from the abscess itself would confirm the infectious agent and potentially identify polymicrobial infections, which are common in renal abscesses. This would strengthen the link between the UTI and the abscess.

Response 6: In our manuscript, only urine culture is reported. As we also mentioned in our manuscript, Siegel, et al. proposed an algorithmic approach to the management of renal abscesses in 1996. They stated that primary conservative management, which involved the use of antibiotics, was advised for small abscesses (< 3 cm), while drainage (percutaneous or surgical) was advised for large abscesses (> 5 cm). In medium-sized abscesses (3-5 cm), both observation protocols were feasible [1]. Nevertheless, percutaneous abscess drainage is associated with a number of uncommon complications, including pyopneumothorax, bacteremia, and fistulas in the gastrointestinal tract.

Bamberger proposed that aggressive interventional or surgical treatment of renal and perinephric abscesses of 5 cm in diameter or less be avoided, as they can achieve complete remission following antibiotic therapy, like in our patient, we achieved complete remission [2]. In an era of enhanced antimicrobial therapy and supportive care, the proportion of conservative treatment relative to therapeutic intervention is expected to rise with time. Nonetheless, the conditions under which medical therapy without drainage is a viable alternative remain ambiguous. A drawback of medical therapy lacking diagnostic drainage is the potential use of empirical regimens without awareness of the responsible organisms and their antimicrobial susceptibilities, as you mentioned in your comments. According to the active guideline recommendations for small renal abscesses (<3cm), no percutaneous or open drainage was necessary as the first intention in our case.

1.Siegel JF, Smith A, Moldwin R. Minimally invasive treatment of renal abscess. J Urol. 1996;155:52–55. [PubMed] [Google Scholar]

2.Bamberger DM. Outcome of medical treatment of bacterial abscesses without therapeutic drainage: review of cases reported in the literature. Clin Infect Dis 1996;23:592-603.

Comments 7: Dapagliflozin Rechallenge: While understandable given the patient's presentation, not rechallenging the patient with dapagliflozin limits the ability to assess whether the drug was indeed the causative agent. A cautious rechallenge under close monitoring, if ethically justifiable and clinically appropriate, could provide further insights.

Response 7: In addition to any existing conditions that may have increased the patient's vulnerability to a UTI resulting in an abscess, it is strongly suggested that SGLT2 inhibitor has played a significant role in the development of the UTI. Unlike women, men typically do not experience UTIs unless there are underlying risk factors such as kidney stones, benign prostatic hyperplasia (adenoma), or urethral strictures that contribute to residual urine retention. Therefore, the presence of these predisposing factors cannot be overlooked when assessing the risk of UTIs in male patients.

Comments 8:Control Group (Lines 181-184): Comparing the incidence of renal abscess in patients on SGLT2 inhibitors with a matched control group (e.g., patients on other heart failure medications) would strengthen the argument for a causal relationship.

Response 8: Other medications that are frequently prescribed for cardiac conditions do not induce MGIs and or UTIs. 

Comments 9:The conclusion that this is the first case of SGLT2 inhibitor-associated renal abscess in a patient without prior risk factors was not fully supported by the evidence presented. The lack of a thorough investigation of potential risk factors weakens this claim. The authors acknowledged the low incidence of UTIs related to SGLT2 inhibitors but did not discuss the even rarer occurrence of renal abscesses. The discussion should contextualize the case within the existing literature on SGLT2 inhibitor-associated infections and acknowledge the limitations of a single case report.

Response 9: This case is particularly significant due to its connection with SGLT2 inhibitor use in a patient suffering from heart failure with reduced ejection fraction, who does not have diabetes, immunosuppression, or any urinary pathologies.
Managing renal abscesses in patients on SGLT2 inhibitors demands careful attention to the specific risks associated with these medications.

Comments 10: Table 1: Provides useful echocardiographic data. However, including baseline echocardiographic parameters before dapagliflozin initiation would allow for assessment of the drug's impact on cardiac function and potentially identify any pre-existing abnormalities.

Response 10: In table 1 that provides echocardiographic parameters there are no changes in parametres after 2 months of treatment with Dapagliflozin. It is a short time for a big drug impact on echocardiographic parametres.

Comments 11: Table 2: Presents lab values with reference ranges, which is helpful. However, including baseline lab values for comparison would strengthen the manuscript.

Response 11:  Table 2 includes baselinse lab values that was performed at admission.

Comments 12: Figure 1: Clearly demonstrates the renal abscess. Adding a follow-up ultrasound image alongside the CT scan in Figure 2 would improve visualization of the resolution.

Response 12: For a better resolution of the renal abcess we considered appropiate to use the same investigation. CT imaging is more accurate and specific than ultrasound, with a diagnostic rate of 96%. We consider this technique can provide better information on abscess size and is better at evaluating perinephric involvement, including the extension of the abscess.

We have carefully revised our manuscript in response to your insightful comments, enhancing its clarity and overall quality. Thank you for your valuable feedback.

Round 2

Reviewer 2 Report

Comments and Suggestions for Authors

The Authors have made corrections according to the Reviewer's suggestions and improved the manuscript.

Reviewer 3 Report

Comments and Suggestions for Authors

Glad with changes